# Nourishing Conversations: Using Motivational Interviewing in a Community Teaching Kitchen to Promote Healthy Eating via a Food as Medicine Intervention

**DOI:** 10.3390/nu16070960

**Published:** 2024-03-27

**Authors:** Sara Temelkova, Saria Lofton, Elaine Lo, Jeannine Wise, Edwin K. McDonald

**Affiliations:** 1Department of Medicine, University of Chicago, Chicago, IL 60637, USA; 2College of Nursing, University of Illinois at Chicago, Chicago, IL 60612, USA; 3Department of Public Health Sciences, University of Chicago, Chicago, IL 60637, USA; 4Good Food Catalyst, Chicago, IL 60612, USA

**Keywords:** Food as Medicine, Culinary Medicine, motivational interviewing, teaching kitchen, food insecurity

## Abstract

It is well known that dietary choices impact both individual and global health. However, there are numerous challenges at the personal and systemic level to fostering sustainable healthy eating patterns. There is a need for innovative ways to navigate these barriers. Food as Medicine (FM) and Culinary Medicine (CM) are approaches to helping individuals achieve healthier diets that also recognize the potential to alleviate the burden of chronic diseases through healthy eating. Teaching kitchens, which offer an interactive environment for learning nutrition and cooking skills, are valuable educational tools for FM and CM interventions. Motivational interviewing (MI), a type of person-centered counseling, facilitates behavior change and may enhance FM and CM programs involving teaching kitchens. In this commentary, we share our experience with using MI in a community-based CM program at a teaching kitchen. In demonstrating our application of MI principles, we hope to offer an additional strategy for improving dietary quality and delivering nutrition education.

## 1. Introduction

A suboptimal diet, one rich in sodium and low in fruits and vegetables, contributes to more deaths globally than any other risk factor for non-communicable diseases, such as cardiovascular disease and diabetes [1]. Improving dietary quality may prevent an estimated one in every five deaths globally [1]. Based on prospective data from the UK Biobank cohort, adopting a healthy diet with more whole grains, nuts, and fruit, as well as less sugary beverages and processed red meat, may increase life expectancy by approximately 10 years for 40-year-olds [2]. Similarly, an analysis of data from several large cohort studies in the United States positioned a high-quality diet as one of the five key lifestyle variables for extending life expectancy in the US [3]. Accordingly, private and federal organizations have allocated funds to make diets healthier in the US [4]. Despite the evidence supporting healthy diets, the question remains: how do you improve dietary behaviors?

The factors underlying dietary choices and behaviors are both varied and complex [5]. The socio-ecological model conceptualizes food acquisition, preparation, and intake as an interplay of “intrapersonal, interpersonal, organizational, environmental, and public policy factors” [6]. Consistent with this framework, the experience of food insecurity, living in neighborhoods shaped by food apartheid, and living in communities that have few or no grocery stores can negatively impact the consumption of nutritious food and leave individuals with poor dietary options [7,8,9]. Conversely, people who live near grocery stores may be more likely to buy produce [10,11,12]. However, solely addressing structural inequities, such as opening new grocery stores, may not necessarily change the quality of diet for many people [13,14,15,16]. An analysis of purchasing habits from 60,000 households found that improving access would only “reduce nutritional inequality by 9%, with the remaining 91% driven by differences in demand” [17]. Intrapersonal factors such as taste preferences, nutritional knowledge, cooking skills, and cultural influences may account for “differences in demand.” How can Food as Medicine (FM) and Culinary Medicine (CM) interventions address intrapersonal factors in attempts to facilitate healthy dietary behaviors?

In this commentary, we highlight the potential role of motivational interviewing (MI) in FM and CM by reviewing interventions using MI as a tool for behavior change in individual and group settings and by sharing our approach to adopting MI in our community-based CM program called Good Food is Good Medicine (GFGM).

## 2. Food as Medicine (FM) and Culinary Medicine (CM)

FM and CM are strategic approaches to promoting healthy eating; however, they both lack standardized definitions [18,19]. Broadly, FM is the idea that food has potential as a therapeutic tool [18]. The American Heart Association (AHA) defines FM more precisely as “the provision of healthy food to prevent, manage, or treat specific clinical conditions in a way that is integrated with the health care sector” [20]. Within this definition, examples of FM interventions include medically tailored meals, medically tailored groceries, and produce prescription programs [4,21]. FM can simultaneously reduce health care costs and address food access gaps by providing healthy food at reduced or no cost [21]. For instance, a microsimulation modeling study found that a 30% subsidy for healthy foods for Medicare and Medicaid recipients could save as much as USD 100 billion in health care costs [22].

Although the provision of food is valuable, a systematic review of FM interventions geared towards fostering fruit and vegetable consumption could not make reliable conclusions about the effectiveness of FM due to the heterogeneity of the interventions [23]. Nevertheless, the authors noted that almost all the studies included a form of nutritional counseling, with several studies utilizing cooking classes or demonstrations as educational tools [23]. The importance of nutritional counseling in FM should not be underestimated given the complexity of determinants of dietary behaviors. Furthermore, the use of cooking classes with FM represents the intersection of FM and CM.

CM integrates the principles of FM with the practical application of hands-on cooking skills and techniques [24]. By equipping individuals with evidence-based dietary recommendations and actionable culinary skills, CM empowers them to make informed lifestyle and diet choices [25]. Increasing home cooking is one of CM’s goals since this allows for more control over salt, sugar, and fat than eating ultra-processed foods or eating out [26]. A meta-analysis of CM interventions indicated CM’s potential in improving nutritional knowledge, culinary skills, dietary quality, and attitudes toward cooking [27]. The authors of the study also commented on the need to optimize the content and format of CM programming.

Teaching kitchens provide an excellent setting for implementing CM programming. Eisenberg et al. recently defined teaching kitchens as a venue (either physical or virtual) “where individuals come together to learn life-enhancing and health-promoting skills, knowledge and strategies through experiential learning involving food” [28]. A study by Tanumihardjo et al. demonstrated how the concept of the teaching kitchen can integrate into a healthcare system and expand to include social care such as food provision to address food insecurity [25]. Despite these advances, there is a dearth of literature detailing how to develop curricula for teaching kitchens and CM programming.

The Cook-Ed^TM^ is an excellent model that addresses this gap [29]. The model describes eight stages of planning, developing, implementing, and evaluating culinary programming. During the development and implementation phase, the authors recommend employing a pedagogy or behavioral change strategy. Motivational interviewing (MI) is one such strategy for behavior change that has been well-established [30].

## 3. Motivational Interviewing (MI)

MI is a person-centered method of guiding to elicit and strengthen personal motivation for change. Miller et al. originally developed MI to help people with substance abuse disorders, with further developments expanding its utility in facilitating health behavior change [30,31]. The four central principles of MI include reflective listening, developing discrepancy by identifying differences in the patient’s held values and their current behaviors, overcoming or “rolling with” resistance by responding with understanding and empathy, and building the patient’s confidence through self-efficacy [30,31]. There are several MI-specific techniques that reflect the principles of MI. These techniques include agenda setting, avoiding argumentation, monitoring the patient’s readiness to change, reinforcing his/her self-motivational statements, and affirming his/her freedom of choice and self-determination [31]. The “spirit” of these techniques, which focus on partnership, acceptance, compassion, and evocation, makes MI useful in a variety of settings [32,33,34].

Incorporating MI with nutrition education can facilitate diet modification beyond that achieved by standard education [32,33]. For example, in an intervention involving 1011 participants in Black churches, those who received three MI phone calls alongside nutritional education reported a significant increase in fruit and vegetable intake by an average of 1.1 servings per day [32]. Another randomized controlled trial with 175 adult women, guided by trained dietitians and MI principles, saw a 1.2% reduction in dietary fat intake compared to a 1.4% increase in the control group, demonstrating MI’s success in promoting high adherence to dietary change [33].

MI may also help underserved populations navigate food insecurity [34]. A randomized controlled trial conducted in a majority-Black and food-insecure population demonstrated that food pantries utilizing MI increase the self-efficacy of their clients [34]. The participants were given fresh food, on-site nutrition education, individual MI at monthly case management meetings, and referrals to community programs and social services based on their needs and goals. The intervention increased food security, self-efficacy, and self-sufficiency [34].

MI has also been adapted for use in groups [31,35]. The four phases of group MI are group engagement, perspective exploration, perspective expansion, and action [35]. One study used MI to address attrition and weight regain in groups receiving behavioral treatment for obesity [36]. A total of 86 patients were enrolled in a 6-week-long MI group program with a 6-month follow-up, carried out by an individual trained in MI and its use in obesity. The intervention helped patients transition to the action or maintenance stages of behavioral changes related to healthy diet and physical activity. The study population demonstrated significant reductions in caloric intake, sedentary time, and body weight, as well as a significant increase in total physical activity.

Although there are few examples of using MI in CM and FM, we were able to implement these techniques in a community-based CM program.

## 4. Good Food Is Good Medicine (GFGM)

Good Food is Good Medicine (GFGM) is a program under the organization Good Food Catalyst that provides weekly cooking and nutrition classes in 6-week cohorts. Classes are held in several under-resourced Chicago neighborhoods, including Englewood, North Lawndale, Garfield Park, and Little Village. The participants are adults and predominantly Black and Latinx. For example, many of our participants come from Garfield Park, where 91.9% of the population is Black and 52.3% has low food access, compared to 21.9% in greater Chicago [37,38]. In general, many live in areas affected by food apartheid [39]. Our programming focuses on improving dietary behaviors and navigating social determinants of health, such as food insecurity.

The GFGM curriculum was developed by a chef with a background in trauma-informed care and a chef-trained physician who is board-certified in obesity medicine and trained in MI. Our curriculum is grounded in the Health Belief Model, which poses three factors for health behavior change: the belief in the consequences of a health threat, a concern for one’s health, and the belief that provided health recommendations would help reduce the threat [31,40,41]. Within our curriculum, we use the model to address the risks associated with poor nutrition, the benefits of a healthy diet, and the perceived barriers to adopting healthier eating patterns. To elicit behavior change, we deliver our curriculum in the “spirit of MI”.

## 5. Utilizing MI in GFGM

We utilize MI in GFGM in several ways. Our participants, also referred to as students, have varying levels of preparedness for implementing healthy cooking behaviors, such as selecting the right ingredients, shopping for healthy groceries, and utilizing healthy cooking techniques. To help our students explore and resolve ambivalence towards eating healthier and cooking more frequently, we focus on understanding their situation. We survey our participants prior to starting each cohort to better learn their resources, preferred cuisines, and eating patterns, and we adapt our teaching strategies accordingly. The intake surveys also inquire about openness to trying new foods and flavors, and obstacles to cooking at home.

The initial class of each cohort opens with a discussion of what brought each student to GFGM. This discussion establishes trust and provides the students with agency. It also creates the opportunity for students to share their perspectives and support each other as peers. We repeatedly use agenda setting throughout the 6-week course, especially when reviewing the recipes and nutrition topics for each session. Involving the participants in agenda setting personalizes their experience and fosters interest and student engagement in our classes.

Throughout our classes, we continuously build trust by integrating reflective listening. For example, a participant may pose a question about modifying a recipe. Our chef instructor will reflectively listen, confirm understanding of the participant’s perspective, and share the concern with the class. We also express empathy by affirming students’ experiences, cultures, and identities. As such, we establish an environment where they feel safe to share their concerns without judgment or criticism. Overall, the group setting creates opportunities for peer support and normalizing shared experiences.

After establishing trust with the members of each cohort and exploring their perspectives, we gradually utilize the MI principle of developing discrepancy to help students identify gaps between the goals outlined in their intake surveys and their current behavior. Furthermore, our programming includes “Ask a Doctor” sessions led by a chef- and MI-trained physician. Our physician has the chance to identify dissonance and develop discrepancy while answering participants’ health-related questions.

We also encourage our students to share their plans for reaching their dietary goals amongst their cohort peers. During this stage, we acknowledge and accept any resistance that arises. In line with the MI technique of “roll with resistance,” we avoid argumentation and, instead, highlight each student’s self-efficacy. Whereas traditional interventions place primary responsibility on the expert to advise the recipient on their behavior, MI drives change through establishing collaboration [31]. Correspondingly, we also use MI in GFGM by empowering participants to find their own solutions to achieving healthier diets and cooking based on their self-reported capacities to change their dietary behaviors. We ultimately affirm our students’ abilities to eat healthier, while empathizing with their challenges to adopting healthier diets. Throughout the cohort, we attempt to foster self-efficacy by having the students share successes with implementing positive changes in their diets and cooking skills.

## 6. Conclusions

Dietary behaviors have health implications. FM and CM are approaches designed to help individuals eat healthier. These approaches acknowledge that dietary goals and challenges are personal, and recommendations are not entirely generalizable from one person to another. Dietary change is replete with challenges, and techniques such as MI can assist with adopting healthier dietary habits. We demonstrate the feasibility of incorporating MI into FM and CM with our program, Good Food is Good Medicine. In our experience, MI helps tailor FM and CM programming to individuals even when provided in a group setting. We also found that using MI in the setting of a teaching kitchen facilitates peer support among class participants. Given the paucity of literature on curriculum development and implementation for teaching kitchens, our article is one of the first to provide insight about the benefits of MI in FM and CM. We recommend further clinical trials and observational studies to elucidate the effectiveness and role of MI in FM and CM.

## Data Availability

The data presented in this study are available on request from the corresponding author.

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
