# Peer review of "Nourishing Conversations: Using Motivational Interviewing in a Community Teaching Kitchen to Promote Healthy Eating via a Food as Medicine Intervention"

_nutrients, 2024, doi:10.3390/nu16070960_

Round 1

Reviewer 1 Report

Comments and Suggestions for Authors

Nourishing conversations using motivational interviewing in a community teaching kitchen to promote healthy eating via a food is medicine intervention presents a novel overview of how motivational interviewing and culinary medicine merge to promote behavior change. The text is well written and flows clearly describing motivational interviewing, food is medicine and culinary medicine programs.  The authors describe the good food is good medicine (GFGM) program, along with some programmatic details. It appears, but is not clear that the authors are involved in the GFGM program.  There should be more clarity about the program as it seems to be the main focus of the paper.  To better understand the GFGM program it would help to inform on when the program began and how often the program is given.  Do the attendees meet once a week or month and for how long (6 months or one year)? The authors refer to the attendees as "students." Are these youth or adult attendees? It would be helpful if there were examples of how the surveys results from the students in the program were used to address ambivalence, barriers and expressing empathy during the program.  If this information is provided the reader will be able to better understand the program and perhaps replicate procedures in their institutions.

Author Response

Dear Reviewer, 

Thank you for your feedback. Below are our responses following your comments:

It appears, but is not clear that the authors are involved in the GFGM program. Now addressed throughout section 4, beginning with "We utilize MI in GFGM in several ways." We now specify that these are our participants, also referred to as students, and our classes.

There should be more clarity about the program as it seems to be the main focus of the paper. To better understand the GFGM program it would help to inform on when the program began and how often the program is given. Do the attendees meet once a week or month and for how long (6 months or one year)? The authors refer to the attendees as "students." Are these youth or adult attendees? Now addressed in section 3: "weekly cooking and nutrition classes in 6-week cohorts," "Participants are adults," and "The GFGM curriculum was developed by a chef with a background in trauma-informed care and a chef-trained physician who is board-certified in obesity medicine and trained in MI."

It would be helpful if there were examples of how the surveys results from the students in the program were used to address ambivalence, barriers and expressing empathy during the program. Now addressed in section 4: "We survey our participants prior to starting each cohort to better learn their resources, preferred cuisines, and eating patterns, and we adapt our teaching strategies accordingly. The intake surveys also inquire about openness to trying new foods and flavors, and obstacles to cooking at home," and "After establishing trust with the members of each cohort and exploring their perspectives, we gradually utilize the MI principle of developing discrepancy to help students identify gaps between the goals outlined in their intake surveys and their current behavior."

Reviewer 2 Report

Comments and Suggestions for Authors

An interesting article describing how MI may enhance the Food as Medicine approach, particularly in underserved populations. 

Overall, this seems to be a concept paper, with no data/results.  With no results presented, the paper would be strengthened by making a stronger connection between MI and the history of its use, with what populations, and how MI has already been used in the community setting/in community nutrition.

A stronger connection is also needed to justify the use of MI with culinary kitchens, and the Food as Medicine approach (also given that there is no consistent approach/definition for using Food as Medicine) 

Additionally, the authors could address- how is MI currently being implemented in the current community program- in a group setting?  Or on a one on one basis, and if so with whom?  Is the instructor trained in MI?  Proper use of MI, and as a counseling technique  is historically time- consuming, the authors should address how this is being addressed in the current program. 

The first 2 sentences in the Introduction need citations. 

Comments on the Quality of English Language

Minor editing of English language 

Author Response

Dear Reviewer, 

Thank you for your feedback. Below are our responses following your comments:

The paper would be strengthened by making a stronger connection between MI and the history of its use, with what populations, and how MI has already been used in the community setting/in community nutrition. We agree the paper would benefit from a stronger MI section. Feedback is now addressed in section 3: added sentence describing examples of the "spirit" of MI, a paragraph on MI use in a food-insecure underserved population, and a paragraph on MI used in a group setting - RCT of patients receiving treatment for obesity.

A stronger connection is also needed to justify the use of MI with culinary kitchens, and the Food as Medicine approach (also given that there is no consistent approach/definition for using Food as Medicine). The Food as Medicine and Culinary Medicine sections are now combined into section 2. This section begins with acknowledgement that both FM and CM lack standardized definitions. In this section we also added a more precise definition of FM from the American Heart Association and information about a model that addresses the gap between teaching curricula and CM programming (Cook-Ed).

Additionally, the authors could address- how is MI currently being implemented in the current community program- in a group setting?  Or on a one on one basis, and if so with whom?  Is the instructor trained in MI?  Proper use of MI, and as a counseling technique is historically time- consuming, the authors should address how this is being addressed in the current program. Now addressed in section 4: "weekly cooking and nutrition classes in 6-week cohorts," "Participants are adults," and "The GFGM curriculum was developed by a chef with a background in trauma-informed care and a chef-trained physician who is board-certified in obesity medicine and trained in MI." Section 5 adds to this with "Further, our programming includes 'Ask a Doctor' sessions led by a chef- and MI-trained physician. Our physician has the chance to identify dissonance and develop discrepancy while answering participants’ health-related questions." Section 4 has been edited for clarity and more succinctly explains our methods - how we utilize MI principles in the group setting, which allows us to reach more students at a time than one-on-one MI without compromising the quality of MI delivery. The conclusion was also revised to better reflect the results of our commentary.

The first 2 sentences in the Introduction need citations. The original first 2 sentences have been removed. The Introduction now begins with cited sentences. Citations throughout the paper were reviewed and removed if not relevant to the research described.

Round 2

Reviewer 2 Report

Comments and Suggestions for Authors

Thank you to the authors for addressing reviewer comments/suggestions. 

This commentary on how authors utilized MI with a specific Food as Medicine- Good Food is Good Medicine program will be of interest to readers and is a now stronger discussion about how MI may be an innovative approach to promote behavior change in the Food as Medicine/Culinary Medicine programming space.